# The Distinct Role of the HDL Receptor SR-BI in Cholesterol Homeostasis of Human Placental Arterial and Venous Endothelial Cells

**DOI:** 10.3390/ijms23105364

**Published:** 2022-05-11

**Authors:** Manuela Strahlhofer-Augsten, Carolin Schliefsteiner, Silvija Cvitic, Meekha George, Ingrid Lang-Olip, Birgit Hirschmugl, Gunther Marsche, Uwe Lang, Boris Novakovic, Richard Saffery, Gernot Desoye, Christian Wadsack

**Affiliations:** 1Research Unit, Department of Obstetrics and Gynecology, Medical University of Graz, 8036 Graz, Austria; manuela.augsten@medunigraz.at (M.S.-A.); carolin.schliefsteiner@medunigraz.at (C.S.); silvija.tokic@medunigraz.at (S.C.); birgit.hirschmugl@medunigraz.at (B.H.); gernot.desoye@medunigraz.at (G.D.); 2BioBank Graz, Medical University of Graz, 8036 Graz, Austria; 3Research Unit of Analytical Mass Spectrometry, Cell Biology and Biochemistry of Inborn Errors of Metabolism, Department of Paediatrics and Adolescent Medicine, Medical University of Graz, 8036 Graz, Austria; 4Otto Loewi Research Center, Division of Pathophysiology and Immunology, Medical University of Graz, 8010 Graz, Austria; meekha.george@medunigraz.at; 5Gottfried Schatz Research Center, Divison of Cell Biology, Histology and Embryology, Medical University of Graz, 8036 Graz, Austria; ingrid.lang@medunigraz.at; 6Otto Loewi Research Center, Division of Pharmacology, Medical University of Graz, 8010 Graz, Austria; gunther.marsche@medunigraz.at; 7Molecular Immunity, Infection and Immunity Theme, Murdoch Children’s Research Institute, Parkville, VIC 3052, Australia; boris.novakovic@mcri.edu.au (B.N.); richard.saffery@mcri.edu.au (R.S.)

**Keywords:** human placenta, SR-BI, HDL, endothelium, arterial-venous difference

## Abstract

As opposed to adults, high-density lipoprotein (HDL) is the main cholesterol carrying lipoprotein in fetal circulation. The major HDL receptor, scavenger receptor class B type I (SR-BI), contributes to local cholesterol homeostasis. Arterial endothelial cells (ECA) from human placenta are enriched with cholesterol compared to venous endothelial cells (ECV). Moreover, umbilical venous and arterial plasma cholesterol levels differ markedly. We tested the hypothesis that the uptake of HDL-cholesteryl esters differs between ECA and ECV because of the differential expression of SR-BI. We aimed to identify the key regulators underlying these differences and the functional consequences. Immunohistochemistry was used for visualization of SR-BI in situ. ECA and ECV were isolated from the chorionic plate of human placenta and used for RT-qPCR, Western Blot, and HDL uptake assays with ^3^H- and ^125^I-labeled HDL. DNA was extracted for the methylation profiling of the SR-BI promoter. SR-BI regulation was studied by exposing ECA and ECV to differential oxygen concentrations or shear stress. Our results show elevated SR-BI expression and protein abundance in ECA compared to ECV in situ and in vitro. Immunohistochemistry demonstrated that SR-BI is mainly expressed on the apical side of placental endothelial cells in situ, allowing interaction with mature HDL circulating in the fetal blood. This was functionally linked to a higher increase of selective cholesterol ester uptake from fetal HDL in ECA than in ECV, and resulted in increased cholesterol availability in ECA. SR-BI expression on ECV tended to decrease with shear stress, which, together with heterogeneous immunostaining, suggests that SR-BI expression is locally regulated in the placental vasculature. In addition, hypomethylation of several CpG sites within the SR-BI promoter region might contribute to differential expression of SR-BI between chorionic arteries and veins. Therefore, SR-BI contributes to a local cholesterol homeostasis in ECA and ECV of the human feto-placental vasculature.

## 1. Introduction

Fetal development depends highly on the bioavailability of cholesterol. As a structural membrane component, cholesterol affects the content of other lipids within the membrane [1,2], regulates the propagation of signal transduction events [3,4], serves as a precursor for steroid hormone synthesis [5], and is essential for the activation of sonic hedgehog homolog [6,7], one of the signals governing morphogenesis.

The fetus may cover its own cholesterol demand either by endogenous cholesterol synthesis or through exogenous cholesterol supplies. However, de novo cholesterol synthesis in the fetus [8] is not sufficient and, therefore, maternal cholesterol is required as an additional source. Maternal cholesterol is transported across the placenta [9,10] to the fetal blood. The placenta takes up cholesterol delivered within lipoproteins through receptor-mediated and receptor-independent transport mechanisms [11,12].

As opposed to adults, in fetal circulation, high-density lipoprotein (HDL) is the main cholesterol carrying lipoprotein. It differs from adult HDL by its higher proportion of apolipoprotein (Apo)E [13], but lower proportion of ApoA1 [14]. We demonstrated that ApoE of fetal HDL regulates antioxidative enzymes in human placental endothelial cells (ECs) [14,15]. This effect requires the interaction of HDL with distinct receptors on ECs.

Scavenger receptor class B type I (SR-BI) is the major HDL receptor expressed on the surface of various cell types, including ECs. SR-BI mediates HDL-dependent signaling in ECs [16] by binding a variety of ligands, including native and oxidized lipoproteins [17,18]. The receptor localizes to specialized plasma membrane compartments, i.e., caveolae, and is best known for its role in facilitating the uptake of cholesteryl esters (CE) from HDL into steroidogenic cells [19,20,21]. Of note, several studies have demonstrated differences in SR-BI gene expression between arteries and veins [22,23,24].

Since SR-BI is expressed on the maternal-fetal interface in human first trimester and term villous trophoblasts [25], it is likely involved in the trans-placental transfer of maternal cholesterol to the developing embryo. Cholesterol from maternal circulation is taken up by trophoblasts on their apical side and is effluxed on the basolateral side to the villous stroma [11]. Further steps of cholesterol transport across the placenta have remained elusive. Cholesterol levels in the mother do not directly correlate with those in the fetus, at least in the second half of gestation [26,27].

Currently, the mechanisms of cholesterol homeostasis in the fetus and the feto-placental vasculature are poorly understood. At the feto-placental interface, the ATP-binding cassette transporters ABCA1 and ABCG1 are involved in the efflux of cholesterol by a two-step mechanism. Although SR-BI operates in a bidirectional manner [28], it is not involved in cholesterol efflux despite its presence in human placental endothelial cells [29,30]. However, because of its location on placental endothelial cells, SR-BI is one promising candidate in balancing fetal and feto-placental cholesterol content. Maternal hypercholesterolemia may alter fetal cholesterol homeostasis and is associated with the accumulation of fatty streaks in the fetal vascular system [26,27], thereby imposing a higher risk for the development of atherosclerotic lesions later on in the offspring’s life [31]. This underpins the necessity to study cholesterol transporters at the feto-maternal interface, as they might be interesting targets for early (e.g., nutritional) interventions to improve long-term offspring health.

In the present study, we tested the hypothesis that SR-BI is expressed on the feto-placental endothelium and differentially between human placental arterial (ECA) and venous endothelial cells (ECV). We investigated both HDL association and CE uptake by these cells as a physiological consequence of the different SR-BI expression in these cells. Furthermore, we aimed to test transcriptional regulators which might explain the differential SR-BI levels between ECA and ECV. As atherosclerotic lesions are located in regions of low wall shear stress [32], we further hypothesized that shear stress modulates SR-BI expression and, hence, represents a regulator in the feto-placental vascular system. To our knowledge, this is the first study comparing SR-BI mediated cholesterol uptake into the ECs of arteries and veins within the same human organ.

## 2. Results

### 2.1. SR-BI Protein Is Elevated in ECA Compared to ECV

Immunohistochemistry of a representative cross section from placental tissue demonstrated a clear difference in staining intensity for SR-BI between endothelial cells lining an artery and a vein within the same placental villus (Figure 1A). A more detailed analysis along the entire vascular tree showed a high diversity of SR-BI localization with focal staining in both segments of the vascular tree and in the capillaries of the microvasculature. In order to quantify these expression differences observed in situ, *SR-BI* mRNA and protein levels were measured in primary ECA and ECV cells isolated from human term placenta after their culture under identical conditions. RT-qPCR demonstrated 20% more (*p* = 0.001) SR-BI transcripts in ECA than ECV (Figure 1B). These results were paralleled by 74% higher (*p* < 0.001) SR-BI protein levels in ECA compared to ECV (Figure 1C,D). Fetal liver was used as the positive control for SR-BI expression. The observed differences in SR-BI abundance were especially pronounced between vascular pairs of matched arterial and venous endothelial cells, but also apparent between randomly selected ECA and ECV (data not shown).

### 2.2. SR-BI Is Predominantly Expressed on the Apical Side of Placental Endothelial Cells

Immunohistochemistry demonstrated that SR-BI is predominantly expressed on the apical side of placental endothelial cells in situ, allowing interaction with mature HDL circulating in the fetal blood (Figure 2A). In addition, a weak staining was detectable in the cytoplasm, whereas the basal side of the endothelial cells was devoid of the SR-BI protein. The cryosection was also immunolabelled for the classical endothelial cell marker von Willebrand factor (Figure 2B) to define the area of the endothelium. The granular staining of the cells indicates the typical endothelial-specific structure. An IgG negative control did not show any specific staining (Figure 2C).

### 2.3. SR-BI Promoter Methylation Differs between ECA and ECV

The stability of the *SR-BI* expression differences in ECA and ECV (despite their isolation and culture under the same culture conditions for up to 10 passages) prompted us to test the hypothesis that the expression differences are the result of an epigenetic effect. To this end, we studied DNA methylation across the *SCARB-1* gene promoter using DNA methylation arrays in paired isolations of ECA and ECV from nine individual donors. DNA methylation is a covalent modification of DNA involving the addition of a methyl (-CH3) group to a cytosine, generally in the context of a Cytosine-phosphate-Guanine (CpG) dinucleotide, by a specific set of enzymes [33]. The role of DNA methylation in gene regulation and chromatin structure is context dependent [34], with an increase in DNA methylation at and around CpG Islands (regions of high CG density in promoter regions of genes) associated with gene silencing [35]. Importantly, DNA methylation is extensively remodeled during embryogenesis and differentiation, and therefore plays a role in cell identity [36]. Both cell populations were equally hypomethylated at the CpG Island within the putative gene promoter (Figure 3A). In contrast, two regions, one directly upstream of the promoter and another in the gene body, showed variation between the two cell types, with several CpG sites showing hypomethylation specifically in ECV as compared to ECA. In total, 98 CpG sites were investigated, of which 20 were differentially methylated between ECA and ECV (Figure 3B,C). The functional significance of these regions is unclear; however, the overall methylation pattern hints towards a promoter-enhancer 3D loop, and the differences in methylation, and therefore *SCARB-1* transcription, might contribute to the differences in the SR-BI levels observed between ECA and ECV.

### 2.4. GATA3 Transcription Factor Might Contribute to Differential SR-BI Expression

We further investigated the expression of the transcription factor GATA3 in ECA and ECV. Several binding sites for GATA3 have been identified in the *SCARB-1* promoter region using in silico analysis (http://www.cbrc.jp/research/db/TFSEARCH.html (TESS—Transcription Element Search System) accessed on 30 January 2022. In addition, microarray data from our group comparing gene expression between ECA and ECV (Gomes L. and Desoye G., unpublished) showed higher GATA3 expression in ECA than ECV. Of note, GATA3 action is known to be crucial to endothelial cell function [38,39,40]. We therefore speculated that the GATA3 transcription factor is likely involved in the regulation of SR-BI. Eventually, RT-qPCR showed that GATA3 mRNA was indeed higher expressed in ECA than ECV (Appendix A), but at the protein level this difference was not significant (Appendix A).

### 2.5. LXR Transcription Is Not Involved in SR-BI Regulation between Vascular Beds

In addition to GATA3, 60 transcription factors possibly involved in the regulation of SR-BI expression were identified by an analysis of the SCARB1 gene promotor region. It is hardly possible to investigate each candidate, but we selected genes based on their possible involvement in placental cholesterol homeostasis. Since LXR is a main regulator of SREBP-1c transcriptional activity, the LXR-SREBP-1c pathway might regulate SR-B1 expression. To study the role of SR-B1 in LXR-mediated effects we tested the effect of LXR agonist TO901317 on the cells. Neither on a mRNA level using qRT-PCR (data not shown), nor on a protein level using Western Blot (Appendix A), could we observe an effect on SR-BI regulation upon LXR agonist treatment.

### 2.6. PDZ Domain-Containing Protein PDZK1 Is Equally Expressed in ECA and ECV

Additionally, PDZK1, an adaptor protein binding to SR-BI, was investigated. PDZK1 mediates lipid uptake and SR-BI signaling [41,42]. Although not critical to SR-BI abundance or subcellular localization in endothelial cells, PDZK1 stabilizes SR-BI in the plasma membrane [43] and contributes to its function. Evidence of PDZK1 abundance on placental endothelium has not been tested in the past, to the best of our knowledge, leading us to speculate that PDZK1 on placental endothelium might be involved in the regulation of SR-BI stability and function. While RT-qPCR revealed higher levels of PDZK1 in ECV compared to ECA (Appendix A), these differences were not confirmed on a protein level (Appendix A).

### 2.7. Neither Oxygen nor Shear Stress Regulate SR-BI Protein in Primary Placental Endothelial Cells

In situ, placental ECA and ECV are exposed to different levels of oxygen (higher oxygen levels in ECV than ECA), which may have accounted for the expression differences. In order to test this, both cell types were cultured under 1, 5, 12, and 21% oxygen and SR-B1 mRNA levels were assessed. The results confirmed the higher SR-B1 mRNA levels in ECA than ECV, which were virtually unaffected by changes in ambient oxygen tension (Appendix A).

Besides oxygen, shear stress is also different between arteries and veins. The blood pressure in the umbilical artery (53 mmHg) is significantly higher as compared to the corresponding vein (20 mmHg) [44]. Therefore, we hypothesized that differences in shear stress between arteries and veins may result in differential SR-BI expression. However, we were unable to demonstrate any differences in SR-BI protein abundance as measured by Western Blot between ECA and ECV in response to shear stress (Appendix A).

### 2.8. Selective HDL CE-Uptake Is Impaired in ECV Compared to ECA

We next tested whether differential SR-BI expression affects cell association of HDL (binding, internalization, and degradation) and CE-uptake of ECA and ECV. HDL_3_ binds to SR-BI via its Apo-AI moiety and, subsequently, becomes also partly internalized and degraded [17,45,46].

The cell association studies showed that cell-associated HDL_3_ was 20 ± 6% higher on ECA compared to ECV; this effect was independent of acceptor concentration (Figure 4A). More than twice as much HDL-CE was selectively taken up by ECA compared to ECV. Remarkably, the addition of BLT-1 inhibited CE-uptake in ECA almost to the level of ECV. BLT-1 showed no effect in ECV, suggesting that SR-BI-mediated HDL-CE uptake does not play a significant role in ECV (Figure 4B). This finding strongly argues for additional SR-BI independent pathways capable for selective CE-uptake. The differences in cell association and uptake of HDL_3_ between ECA and ECV parallel the differential SR-BI levels found in whole cell lysates.

## 3. Discussion

Maternal lipoproteins have been studied extensively during human pregnancy [47], but little data are available about the role of fetal lipoproteins and their role at the feto-placental interface. The key finding of the present study is that endothelial cells from chorionic arteries of the human placenta selectively take up more CE from fetal HDL than their venous counterparts because of an elevated SR-BI receptor expression and effectivity.

The differences between arterial and venous endothelial cells have been studied primarily in relation to arterial-venous vasculogenesis and angiogenesis, and a range of molecules and transcription factors involved in these processes have been identified [24,48,49]. We have established expression differences of classical arterial-venous genes between ECA and ECV in previous studies [23]. However, a functional and regulatory description of endothelial cells related to lipid homeostasis has remained elusive. To the best of our knowledge, this is the first study clearly demonstrating a functional distinction resulting from different protein levels of the HDL-receptor SR-BI between ECA and ECV from the same human organ and vascular loop.

One plausible explanation for expression differences between the vascular beds is a distinct local environment of endothelial cells in arteries and veins. The variability in SR-BI expression along the vasculature within the same placenta reflects local variation within the respective microenvironment. Its importance was shown by several studies, but these mostly focused on its role in angiogenesis and tumor progression [50]. However, micro-environmental regulation of SR-BI expression by cytokines in atheromas has also been demonstrated, affecting reverse cholesterol transport from macrophage-derived foam cells [51].

In the present study, we demonstrated heterogeneous histological staining of SR-BI protein along the vascular tree, and the difference in SR-BI expression was substantiated by quantitative in vitro measurements. The difference between ECA and ECV was stable for up to 10 cell passages. Therefore, and because of the similarity between in situ and in vitro results, we can exclude a cell culture-mediated effect on SR-BI expression differences. We suggest that the observed differences between ECA and ECV are intrinsic to each specific cell type, even though cell culture conditions cannot perfectly mimic the in vivo microenvironment of the respective vascular niche.

Given the stability of differential SR-BI expression in ECA and ECV in long-term cultures, as observed here, it is tempting to speculate about underlying epigenetic mechanisms, such as a DNA methylation, specifically silencing the SR-BI promoter. While analyzing the DNA methylation pattern of the SR-BI promoter region, as well as upstream and downstream regions, we observed distinct changes in the methylation of CpGs between ECA and ECV. In both ECA and ECV, the promoter region itself was hypomethylated, i.e., accessible to transcription factors, but no differential methylation of CpGs between ECA and ECV was found. In contrast, defined alterations in methylation at downstream intronic and exonic regions were apparent between ECA and ECV, but their potential role in the associated differential gene expression remains unclear. Methylation changes in such regions, known as ‘distal elements’ or ‘enhancers’, are commonly associated with altered gene expression [52,53,54]; a direct relationship between the two observations could only be fully explored by detailed functional testing in vitro using reporter-based approaches.

Alternative mechanisms underlying gene expression differences are manifold and include changes of other epigenetic marks such as histone modification profile or epigenetic regulation of a transcription factor upstream of SR-BI. In silico analysis of the SR-BI promoter sequence identified binding sites for 60 known transcription factors including GATA3 and LXR. Although we observed lower GATA3 mRNA levels in ECV, these differences were reciprocated, but not significant, on a protein level. Therefore, lower GATA3 in ECV might contribute to lower SR-BI in this vascular niche, but this could not be demonstrated conclusively.

From previous studies we knew that, similar to GATA3, LXR expression is also higher in ECA compared to ECV [55]; we therefore tested if SR-BI was responsive to LXR agonist TO901317 treatment, which is known to induce other HDL receptors such as ABCA1. However, in neither ECA nor ECV did SR-BI mRNA or protein levels change upon agonist stimulation. Therefore, LXR action apparently does not regulate SR-BI in placental endothelium, as it has been shown in other studies before [56].

Furthermore, miRNAs have the potential to regulate target gene expression. A small number of studies have convincingly demonstrated that certain miRNAs target SR-BI expression, thereby modifying cholesterol uptake and reverse cholesterol transport [57,58,59]. This is an important issue for further research but clearly outside the scope of this study.

As a key adapter protein, PDZK1 might regulate SR-BI action in a post-translational manner. We indeed observed elevated PDZK1 mRNA levels in ECV; however, this was not confirmed on protein level. Given the differences observed in ECA and ECV with regard to the selective uptake of CE -HDL by SR-BI, this difference nevertheless might be of functional relevance.

In addition to transcription factors, we studied (micro-) environmental conditions, which might affect SR-BI regulation. In situ endothelial cells of the arterial and venous segment of any vasculature are exposed to different oxygen tensions and a varying extent of shear stress. In the human placenta, the venous branch of the vasculature transports oxygen enriched (pO_2_ ~ 35 mmHg) blood and nutrients to the fetus. The arterial branch transports blood and waste products back from the fetus to the placenta and, therefore, is low in oxygen (pO_2_ ~ 20 mmHg) [60,61]. However, and in contrast to our assumption, different oxygen conditions did not alter SR-BI mRNA expression levels. Although transcripts do not necessarily represent functional protein, these data make it unlikely that SR-BI is regulated by oxygen in placental chorionic vessels.

Physiological differences in shear stress between arteries and veins are the result of pulsatile blood flow throughout the vasculature generated by the cardiac beat of the fetus [62]. The relationship between local mechanical forces and HDL-cholesterol have been described previously [63]. Independent of the usage of the same shear stress (data not shown) or shear stress adapted to correspond to the physiological situation in arteries and veins, we did not observe significant changes in SR-BI protein abundance upon shear stress exposure. Therefore, laminar shear stress does not seem to affect endothelial SR-BI levels.

Strikingly, the higher SR-BI expression in ECA resulted in a higher selective CE-uptake in ECA versus ECV, which is a direct functional consequence. Of note, exposure of the cells to BLT-1, a potent inhibitor of CE-uptake [64], blocked uptake by 50% exclusively in ECA, even though there was SR-BI expression in both cell types. Thus, SR-BI independent mechanisms apparently contribute to selective CE-uptake in ECA and ECV, perhaps similar to those identified in murine macrophages [65]. There are possible candidates which might facilitate CE uptake in addition to SR-BI. The cubilin/megalin complex has been identified as an additional HDL receptor in many tissues; however, its affinity for HDL is lower than that of SR-BI [66], but two studies have demonstrated that neither cubilin nor megalin is present on placental endothelium [67,68]. Therefore, these receptors are unlikely to contribute to CE uptake from HDL in placental endothelial cells. As fetal HDL is enriched in ApoE [69], a particular ApoE receptor, such as LRP8 (LDL receptor related protein 8), might contribute to this difference. LRP8 protein in ECA and ECV was not differently expressed between the two vascular beds (data not shown), making it a less likely candidate. Despite contradictory data on its involvement in HDL metabolism [70,71], we also studied CD36 protein levels between ECA and ECV (data not shown), but did not observe any difference between the two cell types.

The higher cholesterol uptake from the arterial blood coming from the fetus as compared to the venous blood leaving the placenta has various physiological implications. ECA contain about 1.6-fold more cholesterol than their counterparts [55], and the present data implicate that SR-BI levels are rather regulated in response to these differences. Moreover, the uptake and sequestration of cholesterol from modified lipoproteins in fetal circulation may be a mechanism to protect the placental vascular system. It is noteworthy that, so far, no atherosclerosis or deposition of atherosclerotic plaques have been described in the feto-placental vasculature [72], although they were found in a variety of fetal aortas [26,27], and also in the maternal vessels of the placental bed [73]. A possible explanation is the efficient mechanisms exerted to maintain local cholesterol homeostasis at the feto-placental interface. SR-BI may very well be part of these mechanisms, but other transporters and enzymes such as ABCA1, ABCG1, and PLTP [29,30,55] have also been shown to contribute to this tightly regulated balance of cellular cholesterol. Of relevance, all these proteins are expressed higher in ECA compared to ECV in the human placenta.

Higher placental SR-BI expression in ECAs may also result in enhanced uptake of cholesterol, which is a potent LXR activator and, hence, regulate LXR target genes, too. The higher cholesterol content and higher LXR expression in ECA than ECV [55] strongly supports this suggestion and allows us to conclude another paracrine function from our results: SR-BI takes up fetal HDL-derived cholesterol/oxysterols, which in turn regulate a range of LXR-target genes. In this way, the fetus could regulate specific functions (e.g., cholesterol efflux) in its own tissue, placenta. This concept would also help explain the higher levels of ABCA1, ABCG1, and PLTP in ECA than ECV. Furthermore, since SR-BI is also a multiligand receptor for oxidized LDL, acetylated LDL, and small unilamellar vesicles, its increased expression on ECA may exhibit a clearing function. Metabolic derivatives present in placental arteries may jeopardize the integrity of the endothelial cell layer. In this scenario, fetal HDL may represent a feed-back defense system, a notion supported by our previous finding that it regulates metallothioneins, the enzymes involved in antioxidative defense [15]. Moreover, HDL can counteract the disorganization of membrane lipids in damaged/dysfunctional endothelium in an SR-BI dependent manner [74].

In summary, our study clearly demonstrates that SR-BI levels in placental ECA and ECV are closely associated with the cellular capacity to selectively take up CE from HDL, therefore influencing cholesterol homoeostasis. Nevertheless, SR-BI does not appear to be solely responsible for CE uptake, as SR-BI inhibitor treatment did not completely abrogate CE uptake. Which factors regulate differential SR-BI levels in the placental vasculature, and which other HDL receptors might contribute to cellular cholesterol homeostasis, remains a subject of future studies.

## 4. Materials and Methods

### 4.1. Materials

Antibodies for ß-Actin, von Willebrand factor (vWF), GAPDH, and PDZK1 were purchased from Abcam (Cambridge, UK). Goat-anti-rabbit and goat-anti-mouse negative control antibodies were obtained from Biorad (Hercules, CA, USA). SR-BI was detected using a sequence specific SR-BI antibody purchased from Abcam (Cambridge, UK). For double-fluorescence immunohistochemistry monoclonal mouse-anti-human Desmin, mouse IgG1, and normal rabbit immunoglobulin fraction were obtained from Dako (Glostrup, Denmark). Goat-anti-mouse CyTM3 and donkey-anti-rabbit CyTM2 were purchased from Jackson Immunoresearch, Dianova (Hamburg, Germany). The polyclonal rabbit-anti-human SR-BI was obtained from Abcam (Cambridge, UK). Primers for SR-BI and the ribosomal protein L30 gene were designed using Primer3 software (http://frodo.wi.mit.edu/ (accessed on 19 October 2012; newer version available https://bioinfo.ut.ee/primer3/)) and were purchased from Ingenetix (Vienna, Austria). All primers were chosen to span exons to avoid the amplification of traces of contaminating genomic DNA.

### 4.2. Immunohistochemistry

Cryosections (5 µm) of placental term tissue were air-dried for at least 4 h and stored frozen at −20 °C. Prior to immunostaining, tissue sections were fixed in acetone for 5 min and rehydrated with phosphate buffered saline (PBS). The antibodies and respective IgG controls were diluted in antibody diluent (Dako, Glostrup, Denmark). On control slides, the primary antiserum was replaced by either mouse- or rabbit-IgG fraction (Dako), used in the same concentration as the respective antibodies, or by antibody diluent.

### 4.3. Double-Fluorescence Microscopy

Placental tissue sections were immune-labelled against Desmin (Dako, 2.5 µg/mL, mouse anti-human, monoclonal) for 30 min at room temperature. After washing in PBS, the slides were incubated with CyTM3 (Jackson Immunoresearch, Dianova, Hamburg, Germany, 14 µg/mL, goat-anti-mouse polyclonal) for 30 min, washed again in PBS and exposed to SR-BI (Abcam, Cambridge, UK, 10 µg/mL, rabbit anti-human, polyclonal). Subsequent washing was followed by incubation with CyTM2 (Jackson Immunoresearch, 15 µg/mL, donkey-anti-rabbit, polyclonal) for 30 min. The slides were washed again in PBS, mounted in Mowiol (Hoechst, Frankfurt, Germany), and examined using a Zeiss Axiophot microscope (Oberkochen, Germany).

### 4.4. Light Microscopy

Slides were immune-labelled using the UltraVision LP Detection System (Thermo Scientific, Fremont, CA, USA) according to the manufacturer’s instructions. SR-BI (Novus Biologicals, Littleton, USA, 2 µg/mL, rabbit anti-human, polyclonal) or vWF (Dako, 0.725 µg/mL, immunoglobulin fraction, rabbit anti-human, polyclonal) were applied for 30 min at room temperature. After three washings in PBS-T, slides were incubated with primary antibody enhancer for 10 min, followed by HRP-Polymer for 15 min. The slides were washed again three times in PBS-T, and immune-labelling was visualized by a 5 min exposure to 3-amino-9-ethylcarbacole (ECA, all from UltraVision kit, Thermo Scientific). The slides were counterstained with Mayer’s hemalum (Merck, Darmstadt, Germany) and after washing in distilled water mounted with Kaiser’s glycerol gelatin (Merck).

### 4.5. Isolation of Human Feto-Placental Endothelial Cells from Arteries and Veins

Feto-placental arteries and veins of the chorionic plate were isolated, as previously described (Lang et al.). In brief, arteries and veins of the chorionic plate suitable for isolation (with respect to vessel size and potential branching points) were identified and dissected. Excess surrounding tissue was removed and vessels were washed briefly in cold PBS. Thereafter, vessels were cannulated and fixed onto cannulas using surgical strings. Cannulated vessels were rinsed using a trypsin-DNase I-digest solution which dissolves endothelial cells from the extracellular matrix. ECs were pelleted, washed, and resuspended in Endothelial Basal Medium (EMB, Clonetics, Cambrex, Walkersville, MD, USA) supplemented with 5% pregnant serum and several growth factors (EGM-MV BulletKit, Clonetics). Cells were plated on gelatin-coated 12-well plates and allowed to attach and grow to confluent monolayers before the first sub-passage. Informed consents from all mothers donating placentae for this study were obtained. The isolation protocol is approved by the local ethics board of the Medical University of Graz (29-319 ex16/17).

### 4.6. Cell Culture Experiments

Cells were cultured in endothelial basal medium (EBM, Lonza Clonetics, Cambrex, Walkersville, MD, USA) supplemented with 5% FCS and suitable growth factors (EGM-MV BulletKit, Lonza Clonetics). Only ECA and ECV between passages 5 and 10 were used in the experiments. To determine SR-BI expression levels, cells were seeded on gelatin-coated 12-well plates (10^5^ cells/well) and grown to 90% confluence for 48 h. Prior to isolation of protein or RNA the cells were washed with ice cold HBSS (1x, Gibco).

For TO901317 treatment, cells were grown into confluent monolayers in complete endothelial basal medium (Promocell, Heidelberg, Germany) and incubated with either TO901317 at 2 uM final concentration or an appropriate vehicle control (DMSO) for 24 h at 37 °C and 21% oxygen.

To measure the influence of different oxygen concentrations on SR-BI expression levels, 3.5 × 10^5^ cells were seeded in 25 cm² flasks and cultured for 6 h and 72 h under 1%, 5%, 12%, and 21% oxygen in a hypoxia bench (BioSpherix, New York, NY, USA).

### 4.7. Simulation of Shear Stress

For shear stress experiments, HEPES-buffered solution containing 10 mg/mL bovine serum albumin (BSA) to enhance viscosity was used. Shear stress was applied on an orbital shaker (Heidolph Unimax 1010, IL, USA) for 2 h and 4 h at 37 °C, respectively. This technique induces similar changes in the alignment and the shape of the cells, that are described with the cone-plate viscometer [75]. The mean shear stress (F), to which the cells were exposed, was calculated according to the following equation:F=ηS2rΠf / Δh

η refers to the dynamic viscosity, S to the area of the well, f to the frequency of shaker-rotation, r to the radius of each dish, and h to the height of liquid. Due to the limitations of the technique used, shear stress could not be calculated properly for the entire dish. Thus, shear stress was calculated for cells at the outer edge of the dish. To determine SR-BI expression after shear stress stimulation, cells were seeded on 6-well plastic dishes (2 × 10^5^ cells/well) and grown over night to reach 70–80% confluence. Cells were washed with HBSS and exposed to shear stress (F ≈ 33 and 16 dyn/cm^2^ for ECA and ECV, respectively) in HBSS + 1% BSA for 2 h and 4 h at 37 °C, respectively. Thereafter, cells were washed and lysed. SR-BI was detected by immunoblot analyses, as described below.

### 4.8. Immunoblot Analyses

Sample lysates (10 µg/mL protein) were separated on 10% SDS gels (PreciseTM Protein Gels, Pierce, Rockford, IL, USA) and transferred to nitrocellulose membranes (Biorad, Vienna, Austria). The membranes were stained with Ponceau S (Sigma, Aldrich, St. Louis, MO, USA) to control the blotting efficiency, blocked with 5% non-fat dry milk (BioRad), and immune-probed against SR-BI (1:1000) or PDZK1 (1:1000) over night, or against ß-Actin as loading control (1:25.000 to 1:30.000) for 45 min, prior to several washing steps and incubation with secondary antibody goat-anti-rabbit IgG (1:500–1:1000). Immunoreactive proteins were visualized with ECL chemiluminescence detection reagents (Amersham). Blots were scanned for optical density using the Alpha EaseFC software (Alpha Innotech, CA, USA).

### 4.9. Quantitative RT-PCR

RNA was extracted from tissues and isolated cells with TRI-REAGENT (Molecular Research Center, Cincinnati, OH, USA). RNA concentration and purity was measured in a BioPhotometer (Eppendorf, Hamburg, Germany). For RT-qPCR, 2 µg RNA was reverse-transcribed using a SuperScript ^TM^II Reverse Transcriptase (Invitrogen, Carlsbad, CA, USA). Taqman probes specific to human SR-BI (Hs00194092_A1) and RPL30 (Hs00265497_A1) were obtained from Applied Biosystems (ThermoScientific, Waltham, MA, USA). The expression level of the mRNA was normalized to the relative ratio of the expression of RPL30. The ∆∆CT method was used to calculate the relative SR-BI expression with RLP30 as the endogenous control gene.

### 4.10. Genome-Scale DNA Methylation Analysis: Data Acquisition and Processing

DNA (1 µg) isolated from 9 ECA and 9 ECV cell populations was bisulphite converted using the MethylEasy ^TM^ bisulphite modification kit (Human Genetic Signatures, Sydney, Australia), according to the manufacturer’s instructions. Conversion efficiency was assessed by bisulphite-specific PCR. Hybridization of bisulphite-treated samples to Illumina Infinium Human Methylation450 (HM450) Beadchips was performed at the Australian Genome Research Facility (AGRF, Melbourne, Australia). Raw data files were exported from Genome Studio (Illumina, San Diego, CA, USA) into the R statistical environment (http://cran.r-project.org/index.html accessed on 30 January 2022). Infinium HM450 data was normalized using the SWAN method [76] from the minfi package available from Bioconductor 35. M-values were calculated after removing probes on the sex chromosomes to eliminate any potential gender bias and any poor performing probes, defined as those with a detection *p*-value cut-off > 0.05 for all samples. β-values were derived from intensities as defined by the ratio of methylated to unmethylated probes given by $\beta \;=M\;/\;\left(U+M+100\right)\;$ and were used as a measure of effect size.

### 4.11. HDL-Isolation and Labelling

Fetal blood was collected from the umbilical vein after delivery from 10 uncomplicated pregnancies. High density lipoprotein (HDL) was isolated from pooled plasma by discontinuous ultracentrifugation as previously described [77].

After measuring protein concentration using the Bradford method, HDL_3_ protein was iodinated with ^125^I-Na (NEN, Vienna, Austria) using N-Br-succinimide as the coupling agent [78]. This procedure resulted in specific activities between 50 and 100 d.p.m.ng^−1^ protein with less than 3% lipid-associated activity. All ^125^I-HDL-preparations were controlled by SDS/PAGE to ensure that the preparations were free of radiolytic or oxidative damage. HDL_3_ lipid moiety was labelled with cholesteryl-1,2,6,7-^3^H-palmitate (^3^H-CE, NEN, Vienna, Austria), as previously described [79].

### 4.12. HDL Lipid-Uptake

To determine association of HDL with the cells as a total of cell surface-bound, internalized and degraded HDL_3_, ECA, and ECV were incubated with serum-free Endothelial Basal Medium (EBM, Lonza), and increasing amounts of ^125^I-labeled HDL_3_ (5, 10, 20 µg protein/mL) were added. Selective CE-uptake of HDL was measured as previously described [80]. BLT-1 potently blocks CE-uptake by SR-BI [64]. Therefore, in parallel, experiments in which the SR-BI inhibitor BLT-1 (10 µM) was added together with HDL were carried out as described above.

### 4.13. Statistical Analysis

The results are presented as mean ± standard error of the mean. After testing for normal distribution (Kolmogorow-Smirnow-Test), either the Student’s *t*-test or Mann-Whitney test was used to determine significance levels of the differences. If two or more groups were compared, ordinary one-way or two-way ANOVA was applied as appropriate. GraphPad Prism v8.0 was used for calculations and preparation of graphs. *p* < 0.05 was accepted as statistically significant.

## Figures and Tables

**Figure 1 ijms-23-05364-f001:**
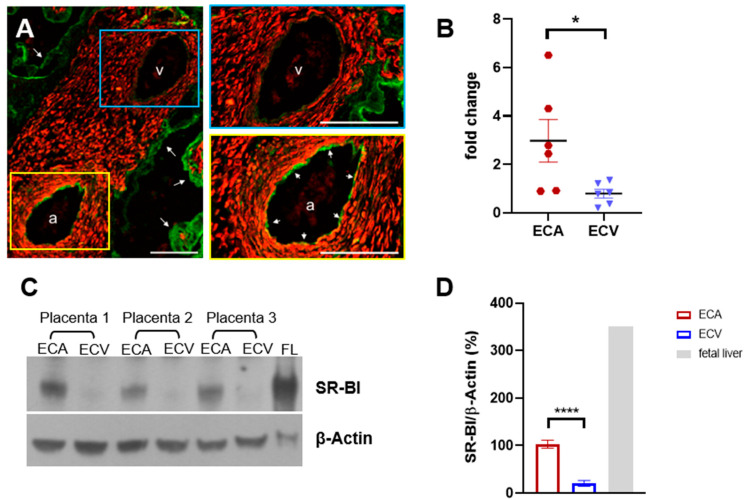
(**A**) Immune fluorescence staining against SR-BI in a term placental stem villus. SR-BI (green) was found on the syncytiotrophoblast (white arrows, overview picture) and on the endothelium of villous arteries (yellow framed detail picture) but not on the endothelium of villous veins (blue framed detail picture). Placental tissue was double-stained against Desmin (red) to visualize villous structures. Scale bars: 80 µm. (**B**) qRT-PCR demonstrated higher SR-BI mRNA expression in ECA compared to ECV (*n* = 6 each, mean ± SD, *t*-test). (**C**) Representative Western Blot of SR-BI protein in ECA and ECV from three different placentae. β-Actin was used as the loading control. (**D**) Densitometric quantification of SR-BI relative to β-Actin as detected by Western Blot in paired ECA and ECV (*n* = 6), fetal liver served as a positive control for SR-BI detection. ECA showed higher SR-BI protein abundance compared to their respective venous counterparts (mean ± SD, one-way ANOVA), * *p* < 0.05, **** *p* < 0.0001.

**Figure 2 ijms-23-05364-f002:**
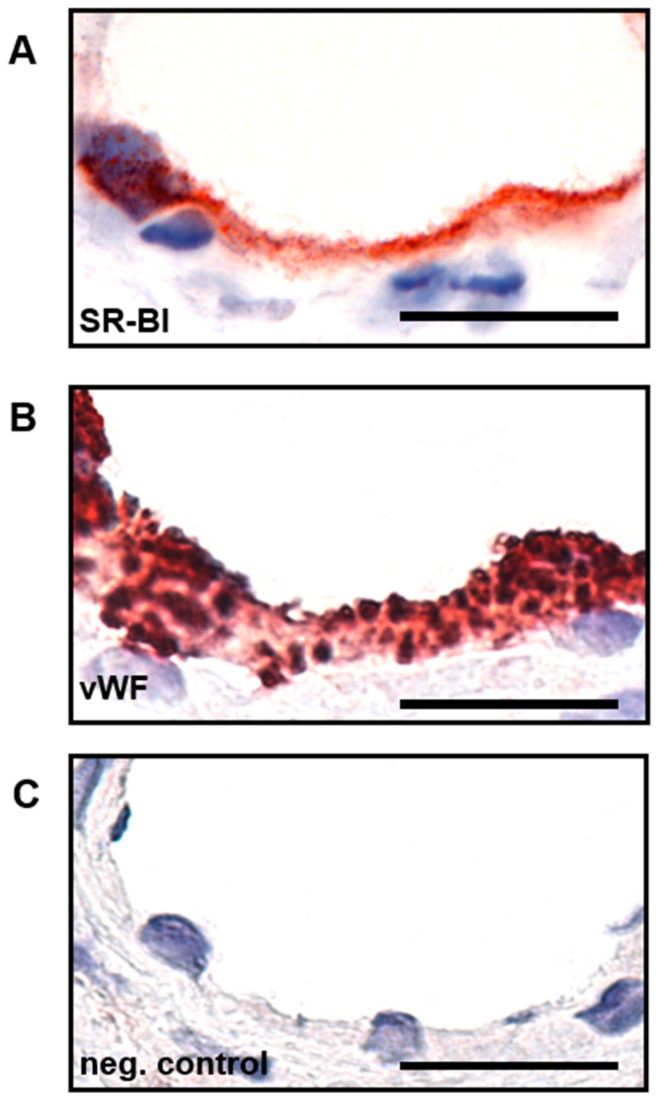
(**A**) Immunohistochemical localization of SR-BI on placental vessels. SR-BI staining (red) is strong on the apical side of the endothelium, whereas in the cytoplasm only a weak signal is detectable. SR-BI is not present on the basal side. (**B**) von Willebrand factor marked EC layer. (**C**) IgG negative control; Bar: 20 µm.

**Figure 3 ijms-23-05364-f003:**
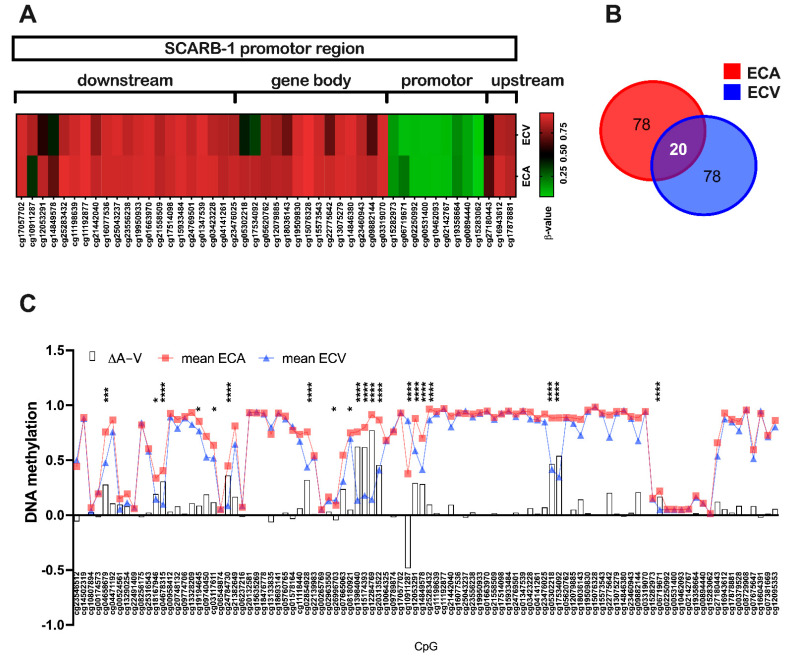
(**A**) Schematic *SCARB1* promoter region structure and related differential CpG methylation pattern between ECA and ECV (*n* = 9, paired). The heatmap uses β-values as the measure of DNA methylation. A total of 98 CpGs were investigated. (**B**) Venn diagram representing differentially regulated CpG islets between ECA and ECV. (**C**) Differentially methylated CpG islets between ECA and ECV; of 98 investigated islets, 20 were significantly different between ECA and ECV. Of these 20, only one CpG islets was hypermethylated in ECV compared to ECA. In the remaining 19 CpGs, methylation was higher in ECA than ECV. In addition to the degrees of methylation (β-values) of ECA (red line) and ECV (blue line), the difference ΔA-V is given as well as white bars. Statistical significance was calculated using M-values instead of β-values, as these are more robust [37]; two-way ANOVA with Sidak’s post hoc test to adjust for multiple comparisons was used, * *p* < 0.05, *** *p* < 0.001, **** *p* < 0.0001.

**Figure 4 ijms-23-05364-f004:**
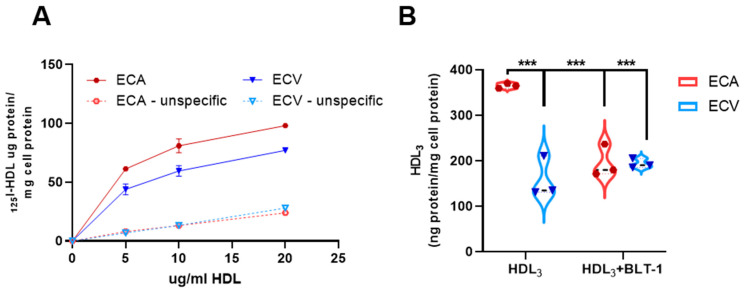
(**A**) Specific binding of increasing amounts of ^125^I-labelled HDL to ECA (solid red line) and ECV (solid blue line) at 37 °C was investigated and showed a higher binding of HDL to ECA than ECV. Unspecific binding (dashed lines) at 37 °C in excess of unlabeled HDL is also shown. (**B**) A total of 10 ug/mL of ^3^H-labelled HDL_3_ were offered to ECA and ECV (*n* = 3) in the presence or absence of the SR-BI inhibitor BLT-1. Selective uptake in ECA was increased compared to ECV, and this increase dropped upon treatment with BLT-1. Nevertheless, a residual selective uptake of HDL occurred in ECA. Two-way ANOVA with Sidak’s post-hoc test for multiple comparisons was used, *** *p* < 0.001.

## Data Availability

The genome-wide DNA methylation dataset produced in this study is available at Gene Expression Omnibus GSE106099 (https://www.ncbi.nlm.nih.gov/geo/query/acc.cgi?acc=GSE106099 (accessed on 4 May 2022)).

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
