# Peer review of "The Distinct Role of the HDL Receptor SR-BI in Cholesterol Homeostasis of Human Placental Arterial and Venous Endothelial Cells"

_ijms, 2022, doi:10.3390/ijms23105364_

Round 1
Reviewer 1 Report
Synopsis:
Cholesterol homeostasis in the fetal circulation is maintained differently from that in the adult, and uptake of cholesterol by the fetus remains understudied. Based on the group’s previous discovery, in this study, Strahlhofer-Augsten et al investigated the mechanism of cholesterol accumulation on the endothelial cells from the human arteries (ECA) and veins (ECV). The major finding is the differential expression of SR-BI on ECA and ECV, and the authors used both in vivo and in vitro approaches to illustrate a possible consequence of cholesterol ester uptake from the fetal HDL. The authors also delved into molecular genetics to figure out what might have regulated such differential expression of SR-BI on different tissues. While the presented results did not seem sufficient to explain the differential SR-BI expression, more is expected to come following this initiative. As described below, this reviewer raises some minor concerns for the authors’ attention. When revised, this reviewer believes this will attract a broad audience.
Concerns:
1. The authors specifically looked into GATA3 and PDZK1, but some cholesterol regulating genes were not shown, such as SREBP1. The rationale to focus on these two is not clear to this reviewer.
2. The epigenetic analysis identified 20 islets that may be associated with differential expression of SR-BI in ECA and ECV. While the authors provided brief speculation (lines 149-151), it is unclear to this reviewer how this difference impacts SR-BI expression difference. For example, methylation has been heavily studied, and a number of enzymes or histone-associated proteins are clearly indicated to regulate the extent of DNA methylation. To provide some mechanistic hypotheses, this reviewer suggests elaborating the role of epigenetics in SR-BI expression patterns here.
3. The authors observed the effect of shear stress on mRNA transcripts, but not protein levels. As the authors pointed out that this is not uncommon, it would benefit the readers to appreciate the authors’ perspective and elaborate on the possibility of post-transcriptional or post-translational or other differences.
4. Minor technical concern on Fig 4. The authors carried out cell surface association of HDL at 4°C, but the unspecific controls were done at 37°C. It is not clear to this reviewer whether the control is relevant.
5. At the end of the discussion, the authors brought up the regulation by LXR. Was an LXR agonist used in any of the in vitro experiments? To what extent is LXR related to the distinct expression of SR-BI in ECA and ECV? It is not clear to this reviewer about the role of LXR here.
6. Some English sentences read weird. A couple more proofreading would help.
Author Response
Firstly, we whole-heartedly thank the reviewer for their overall positive perception of the presented manuscript. In the following, we will address their concerns point by point.
Concerns:
1. The authors specifically looked into GATA3 and PDZK1, but some cholesterol regulating genes were not shown, such as SREBP1. The rationale to focus on these two is not clear to this reviewer.
A: The analysis of the upstream SR-BI promotor region identified about 60 transcription factors possibly implied in the regulation of SR-BI expression. In our lab, microarray data on genes differentially expressed between ECAs and ECVs is available (Gomes L., Desoye G., unpublished). Sequences of the 60 identified transcription factors were blasted with these microarray data and GATA3 was identified to be higher expressed in ECA than ECV, making it a specific target for our further investigation of SR-BI regulation. We justified the rationale of our study now more specifically within the manuscript (Lines 195-205).
In polarized cells, the interaction of the C terminus of SR-BI with a protein called CLAMP, or PDZK1, is required for cell surface expression of the receptor there being an critical adaptor protein for SR-BI mediated actions. Presence of PDZK1 on placental endothelium was not investigated in the past; we therefore suggested a differential expression of PDZK1 between ECA and ECV, which was however, not corroborated by our obtained data (See lines 211f).
Given that 60 transcription factors possibly involved in the regulation of SR-BI expression, it is impossible to look at all of them in detail. For this revision, we further investigated LXR involvement (see below). However, we want to stress, that although we could not identify one specific regulator explaining the differences in SR-BI between arterial and venous endothelium, we could determine the functional consequence of this differential expression studying selective cholesterol ester uptake from HDL. Finally, this functionality appears, at least to us, more relevant.
- The epigenetic analysis identified 20 islets that may be associated with differential expression of SR-BI in ECA and ECV. While the authors provided brief speculation (lines 149-151), it is unclear to this reviewer how this difference impacts SR-BI expression difference. For example, methylation has been heavily studied, and a number of enzymes or histone-associated proteins are clearly indicated to regulate the extent of DNA methylation. To provide some mechanistic hypotheses, this reviewer suggests elaborating the role of epigenetics in SR-BI expression patterns here.
A: An additional explanatory paragraph, hopefully elaborating on the role of epigenetics in this context, and citing relevant literature, is now provided (line Lines 163-171).
- The authors observed the effect of shear stress on mRNA transcripts, but not protein levels. As the authors pointed out that this is not uncommon, it would benefit the readers to appreciate the authors’ perspective and elaborate on the possibility of post-transcriptional or post-translational or other differences.
A: Shear stress was in fact investigated on protein level using Western Blot against SR-BI (Figure S1 Panel B). However, the relevance of hypoxia was only investigated on mRNA level and was not followed up further on protein level, as no effect was observed (Figure S1 Panel A). We apologize, if these findings were not emphasized comprehensively enough within the manuscript. We provide clarification now (Lines 229f and figure legend Fig. S1).
- Minor technical concern on Fig 4. The authors carried out cell surface association of HDL at 4°C, but the unspecific controls were done at 37°C. It is not clear to this reviewer whether the control is relevant.
A: We are deeply sorry, but this appears to be an error that slipped into the manuscript. Unspecific binding was investigated using a 40x excess of unlabeled HDL, but in fact at 37°C. The error has been corrected within the manuscript (Figure Legend Figure 4). As selective binding and uptake are calculated from the difference between total and unspecific binding, this control is in fact quite relevant.
- At the end of the discussion, the authors brought up the regulation by LXR. Was an LXR agonist used in any of the in vitro experiments? To what extent is LXR related to the distinct expression of SR-BI in ECA and ECV? It is not clear to this reviewer about the role of LXR here.
A: We now provide new data on TO901317 treatment of ECA and ECV, investigating both mRNA by qRT-PCR and protein levels by Western Blot. However, we did not observe any effect of TO901317 exposure on SR-BI expression, making it an unlikely regulator of SR-BI (see lines 207-217).
- Some English sentences read weird. A couple more proofreading would help.
A: We worked over the entire manuscript and hope that it is now easier and more pleasing to read.
Reviewer 2 Report
In this work, the authors are aimed at evaluating prospective differences in SR-B1 expression between ECA and ECV. The main findings suggest that SR-B1 expression in higher in ECA than ECV, and that this difference may be attributable to a diverse methylation pattern. Despite the interesting topic, the provided experimental evidence is minimal and several concerns undermine the strength of this manuscript.
As an example:
- GATA3 evaluation by mRNA or protein levels is poorly informative. Since the authors speculate a role for this transcription factor, ChIP analysis for GATA3 on SCARB1 gene should be performed.
- The promoter of SR-B1 contains both lxre and SRE, being responsive to the activity of LXR and SREBPs proteins. The involvement of these transcription factors should be analyzed.
- The results provided by the authors suggest that SR-B1 is not involved in the uptake of HDL-CE in ECV. Since CD36 is another known player involved in the uptake of HDL/cholesteryl esters, this receptor should be characterized in this study.
- Data are expressed as the mean ± SEM. The standard error of the mean indicates the uncertainty of how the sample mean represents the population mean. In my opinion, the authors inappropriately report the SEM instead of the Standard Deviation (SD). Since the SEM is always less than the SD, it deceives the reader into underestimating the variability between replicates/individuals within the study sample. Thus, SEM should be substituted by SD.
- In the "Discussion" section, the authors stated that "The most plausible explanation for expression differences is a distinct local environment of endothelial cells in arteries and veins". However, cells lose the stimuli provided by the local environment in cell cultures. Thus, the authors should better justify the validity of the cell culture experiments.
Author Response
In this work, the authors are aimed at evaluating prospective differences in SR-B1 expression between ECA and ECV. The main findings suggest that SR-B1 expression in higher in ECA than ECV, and that this difference may be attributable to a diverse methylation pattern. Despite the interesting topic, the provided experimental evidence is minimal and several concerns undermine the strength of this manuscript.
Firstly, we thank the reviewer for acknowledging that the topic is of interest. We hope, that within the revision process, we could improve the manuscript and provide additional data to strengthen our presented study. We address all mentioned concerns point by point in the following.
As an example:
- GATA3 evaluation by mRNA or protein levels is poorly informative. Since the authors speculate a role for this transcription factor, ChIP analysis for GATA3 on SCARB1 gene should be performed.
A: We are deeply sorry, but starting chromatin immunoprecipitation (ChIP) assays to identify links between the genome and the proteome from scratch, was not possible given the limited time frame for revision, as we don’t have expertise with these kind of assays to build on. Nevertheless, we agree that ChIP would have been a very valid approach to address this issue.
- The promoter of SR-B1 contains both lxre and SRE, being responsive to the activity of LXR and SREBPs proteins. The involvement of these transcription factors should be analyzed.
A: We did now study LXR involvement in SR-BI regulation using TO901317 and investigating SR-BI on mRNA and protein level, but could not observe any changes in SR-BI in response to LXR treatment (see lines 207-217).
Of note, promotor analysis of the SCARB1 gene yielded 60 possible transcription factors which might be involved in the genes regulation, but not all of them can be investigated within the scope of this manuscript. For GATA3 and LXR, preliminary and published data, respectively, were available in our lab, demonstrating that these transcription factors are higher expressed in ECA then ECV, but concluding from our data, unfortunately neither of the two appears to regulate SR-BI in the way we initially suggested. We aimed to better elaborate why specifically GATA3 (Line 199ff) and PDZK1 (Line 222ff) were initially chosen and focused on within the scope of the study and, as mentioned before, added data on LXR (Line 207-217).
Again, given the limited time frame for revision, we focused on LXR involvement but did not study in particular SREBP or ChREBP in SR-BI expression, although these transcription factors of course might play a role in SR-BI regulation, too.
- The results provided by the authors suggest that SR-B1 is not involved in the uptake of HDL-CE in ECV. Since CD36 is another known player involved in the uptake of HDL/cholesteryl esters, this receptor should be characterized in this study.
A: As a matter of fact, involvement of CD36 in HDL metabolism is under discussion and there is contradicting evidence with regard to its role mediating CE-uptake (1–4)*. Given its very unique structure with very short intracellular domains, it might act as a co-receptor, but we don’t believe that CD36 could fully substitute for SR-BI e.g. in ECV.
Nevertheless, we did investigate CD36 protein in ECA and ECV in the past, and did not observe any difference in CD36 abundance between the two. We therefore conclude, that CD36 does not play a relevant role in HDL or CE uptake in addition to SR-BI that would should be characterized in uptake studies. We added a mentioning of these data, but did not show them, in the manuscript (lines 454ff).
- Data are expressed as the mean ± SEM. The standard error of the mean indicates the uncertainty of how the sample mean represents the population mean. In my opinion, the authors inappropriately report the SEM instead of the Standard Deviation (SD). Since the SEM is always less than the SD, it deceives the reader into underestimating the variability between replicates/individuals within the study sample. Thus, SEM should be substituted by SD.
A: As suggested by the reviewer, the data in Figure 1B and Figure 1D, as well as Supplementary Figure S2 have been modified to show mean +/- SD. Figure 4A, in fact, was already presented in mean +/- SD but wrongly stated. Figure 4B uses a violin plot containing the relevant information, as well as the box and whisker plots in Supplementary Figure S1.
- In the "Discussion" section, the authors stated that "The most plausible explanation for expression differences is a distinct local environment of endothelial cells in arteries and veins". However, cells lose the stimuli provided by the local environment in cell cultures. Thus, the authors should better justify the validity of the cell culture experiments.
A: We are aware of the fact that both the isolation process as well as in vitro culture conditions strongly affect cellular behavior as compared to the in vivo situation. We have a long-standing experience in working with placental endothelial cells from both vascular beds and have fully characterized these two cell types, underpinning that they maintain their differential characteristics also under in vitro conditions (5–7). Furthermore, we do show that the levels of SR-BI between arteries and veins in vivo (Figure 1A ) nicely replicate in the in vitro setting (Figure 1B). We also referred to possible effects of culture conditions in the discussion of the original manuscript, but might not have emphasized this enough. We now hopefully clarified this in the revised version (Lines 306-313).
*
- Connelly MA, Klein SM, Azhar S, Abumrad NA, Williams DL. Comparison of class B scavenger receptors, CD36 and scavenger receptor BI (SR-BI), shows that both receptors mediate high density lipoprotein-cholesteryl ester selective uptake but SR-BI exhibits a unique enhancement of cholesteryl ester uptake. J Biol Chem. 1999 Jan;274(1):41–7.
- De Villiers WJS, Cai L, Webb NR, De Beer MC, Van der Westhuyzen DR, De Beer FC. CD36 does not play a direct role in HDL or LDL metabolism. J Lipid Res [Internet]. 2001;42(8):1231–8.
- Brundert M, Heeren J, Bahar-Bayansar M, Ewert A, Moore KJ, Rinninger F. Selective uptake of HDL cholesteryl esters and cholesterol efflux from mouse peritoneal macrophages independent of SR-BI. J Lipid Res [Internet]. 2006 Nov 1;47(11):2408–21.
- Brundert M, Heeren J, Merkel M, Carambia A, Herkel J, Groitl P, et al. Scavenger receptor CD36 mediates uptake of high density lipoproteins in mice and by cultured cells. J Lipid Res. 2011 Apr;52(4):745–58.
- Lang I, Schweizer A, Hiden U, Ghaffari-Tabrizi N, Hagendorfer G, Bilban M, et al. Human fetal placental endothelial cells have a mature arterial and a juvenile venous phenotype with adipogenic and osteogenic differentiation potential. Differentiation [Internet]. 2008;76(10):1031–43.
- Lang I, Pabst MA, Hiden U, Blaschitz A, Dohr G, Hahn T, et al. Heterogeneity of microvascular endothelial cells isolated from human term placenta and macrovascular umbilical vein endothelial cells. Eur J Cell Biol [Internet]. 2003 Apr;82(4):163–73.
- Leopold B, Strutz J, Weiß E, Gindlhuber J, Birner-Gruenberger R, Hackl H, et al. Outgrowth, proliferation, viability, angiogenesis and phenotype of primary human endothelial cells in different purchasable endothelial culture media: feed wisely. Histochem Cell Biol [Internet]. 2019;152(5):377–90.
Round 2
Reviewer 2 Report
In my opinion, the authors could have made a greater effort to enrich their work with experimental evidence with the objective to strengthen their hypotheses. However, they provided a satisfactory point-by-point response letter and adequately justified the criticisms.